# Aggregated Throughput Prediction for Collated Massive Machine-Type Communications in 5G Wireless Networks

**DOI:** 10.3390/s19173651

**Published:** 2019-08-22

**Authors:** Ahmed Adel Aly, Hussein M. ELAttar, Hesham ElBadawy, Wael Abbas

**Affiliations:** 1Department of Basic and Applied Sciences. Arab Academy for Science, Technology and Maritime Transport (AASTMT), Cairo P.O. Box 2033, Egypt; 2Department of Electronics and Communications Engineering. Arab Academy for Science, Technology and Maritime Transport (AASTMT), Cairo P.O. Box 2033, Egypt; 3Network Planning Department, National Telecommunication Institute (NTI), Cairo 11432, Egypt

**Keywords:** 5G, mMTC, IoT, CSMA, SINR, throughput, polynomial interpolation

## Abstract

The demand for extensive data rates in dense-traffic wireless networks has expanded and needs proper controlling schemes. The fifth generation of mobile communications (5G) will accommodate these massive communications, such as massive Machine Type Communications (mMTC), which is considered to be one of its top services. To achieve optimal throughput, which is considered a mandatory quality of service (QoS) metric, the carrier sense multiple access (CSMA) transmission attempt rate needs optimization. As the gradient descent algorithms consume a long time to converge, an approximation technique that distributes a dense global network into local neighborhoods that are less complex than the global ones is presented in this paper. Newton’s method of optimization was used to achieve fast convergence rates, thus, obtaining optimal throughput. The convergence rate depended only on the size of the local networks instead of global dense ones. Additionally, polynomial interpolation was used to estimate the average throughput of the network as a function of the number of nodes and target service rates. Three-dimensional planes of the average throughput were presented to give a profound description to network’s performance. The fast convergence time of the proposed model and its lower complexity are more practical than the previous gradient descent algorithm.

## 1. Introduction

The evolution of the fifth generation of cellular mobile systems (5G) has become one of the most significant fields for commercial applications. The 5G system is promising to increase data rates by 10 times that of the traditional Long-Term Evolution (LTE) networks, to an average of 10 Gbps with a 1 ms round-trip latency. This high bandwidth is to accommodate an enormous number of connected devices per unit area under the Internet of Things (IoT) framework [1]. In fact, the 5G requirement covers a wide range of core services, specifically the massive Machine-Type Communications (mMTC) is one of the top three services. The other core services being the ultra-reliable low latency (URLLC) and the extreme mobile broadband (eMBB) communications [2]. The services in mMTC are defined by large numbers of linked devices that are generally transmit data traffic. It includes algorithms, mechanisms, and techniques that permit the exchange of information or data without explicit human involvement.

Recent research studies have shown that most of the existing machine-type communications suffer from limited coverage and access reservation. Currently, the procedure for reserving access is limited to a low number of devices and each device requires high data rates [3]. The main challenge in mMTC is the need for efficient connectivity for this massive number of devices that share packets of data. Additionally, mMTC suffers from losses in data packets due to heavy traffic and congestions. A proper way to overcome these data losses is to provide suitable quality of service (QoS) requirements such as high network throughput with low latency. 

In [4], an overview of key radio resource management techniques for 5G dense small cells was studied. Preliminary system-level simulation results indicated that a mean throughput gain of around 63% and up to 84% in latency reduction can be achieved by utilizing resource management techniques. In [5], an efficient online scheme was proposed for predicting channel state information from historical data, in 5G wireless communication systems. The experiment results showed that the scheme not only obtained the predicted channel state information values very quickly but also achieved highly accurate predictions with up to 2.650%–3.457% average difference ratio between the prediction and measurements. In [6], a new Machine-to-Machine (M2M) communication paradigm based on cognitive radio technology was studied, namely the cognitive M2M communication. The cognitive M2M network architecture and cognitive machine model is presented and the coexistence of cognitive M2M devices in TV white spaces was discussed. Additionally, a spectrum exploration scheme motivated by energy-efficiency was introduced. Numerical results show important energy savings and efficiency in providing smart grid data transmission. In [7], device-to-device (D2D) energy-efficient resource allocation algorithm was introduced. To enhance QoS efficiency, a distributed interference mitigation mechanism consisting of a method for canceling interference and a method for optimizing transmission power constraint was discussed. Simulations analyze the achievable performance of the proposed algorithm and discuss implementation and complexity. Additionally, intelligent energy management based on the safe transfer of information between millions of sensors and actuators installed with little or no human involvement was developed in [8]. By investigating the inclusion of software-defined networking with machine-to-machine communication, this motivates the study of a coherent communication structure for intelligent energy management. The proposed software-defined machine-to-machine system was described, with a focus on its price reduction, resource allocation, and end-to-end service quality assurance. In [9], two-stage access control and resource allocation algorithm were developed. In the first phase, a contract-based incentive system was introduced to motivate some delay-tolerant machine-type communication equipment. A long-term cross-layer online resource allocation method was suggested in the second phase, which optimized rate control, energy allocation, and channel choice, without previous channel state information. Finally, under different simulation situations, the performance of the suggested algorithm was verified.

On the other hand, optimizations regarding carrier sense multiple access with collision avoidance (CSMA/CA) have met with great success in different applications. Research on CSMA/CA has a long tradition for years, on which a node senses the channel before transmitting on a shared transmission medium to avoid collision of data in wireless networks [10]. Recent papers [11,12,13,14] proposed various CSMA/CA scheduling algorithms that are able to optimize network QoS metrics, particularly the network throughput. In [11], a CSMA scheme was formulated in which throughput and power consumption of each node were optimized by controlling back-off and sleeping timers, while ensuring throughput optimality. In [12], link throughput was analyzed by taking back-off collisions into account; a model was formed to characterize the collision effect among the network’s nodes. Results showed that their model was robust against different network topologies. In [13], the performance of CSMA network’s throughput was studied under the signal to interference and noise ratio (SINR) model, where a packet was received as long as a certain SINR threshold was exceeded. In [14], they provided effective carrier sensing threshold adjustment algorithms for large wireless CSMA networks. Simulation for evaluating consistency and goodput guaranteed safe interference. They also introduced dynamic signal detection thresholds depending on neighboring transmission feedback. In [15], a distributed iterative algorithm was studied, which produced approximate solutions motivated by an approximation that allowed the expression of approximate solutions via a certain non-linear system with a polynomial size. Numerical results showed that the algorithm produced highly accurate solutions and converged much faster than previous studies. In [16], a distributed scheduling algorithm for the SINR model was studied and proved to be throughput optimal. Further, the algorithm was augmented by using a parallel update technique and the numerical results showed a good performance in terms of a supportable throughput and the convergence rate to steady-state. Moreover, a random access channel evaluation and load estimation of a large number of MTC devices were developed in [17]. A closed-form expression and an effective approach for achieving the Joint Probability Distribution Function (PDF) were extracted from the amount of effective and collided access requests within a random access opportunity. Numerical results justified their formulation’s efficiency and demonstrated that the computational cost was smaller than that of other similar techniques.

There has been extensive consideration to the issue of connection scheduling for peak throughput performance with a focus on the maximum weight scheduling model established in [18,19]. Despite its optimization characteristic, a central controller is needed. In addition, solving for each schedule choice is a non-deterministic polynomial-time hard (NP-hard) problem. Various studies have tried to modify the algorithm of maximal weight to make it easier to deploy [20,21,22]. Such methods are greedy and might not, however, attain an optimized performance. A series of articles [23,24,25] optimized the throughput calculations for a group of connection scheduling schemes called adaptive CSMA, which can, thus, maintain any attainable rate. In particular, the transmission attempt rate was tuned by each connection to guarantee adequate average desired rates of service.

In [26], the transmission attempt rate was adjusted in the CSMA algorithm to support the required target service rates. This technique poses a problem in adjusting the transmission attempt rate parameter, as it is an NP-hard problem, which is difficult to handle. In addition, most of the research in this field aims at solving this problem using a stochastic gradient descent algorithm which is an iterative optimization method for differentiable objective functions [23]. The drawback of this method is that it consumes millions of iterations to converge. Unfortunately, this approach results in an impractical time of convergence depending on the size of the network. Few studies focus on using a proper SINR model for interference, as most of the studies just settled with using interference model based on conflict graph [23,24,25].

The work in this paper aimed to enhance the performance of global dense networks with a CSMA scheduling algorithm under a more practical and realistic SINR model, to adequately capture the complexity of wireless network interferences. In addition, by utilizing an approximation technique to overcome large network sizes, the optimization problem could be solved as the approximation technique distributes large networks into smaller ones. The size of the network and node distance with its neighbors is independent of the size of the whole large network. Thus, the solution to such optimization is achievable due to the scale of the small networks. This means that the global optimization function of the transmission attempt rate parameter in CSMA is distributed into local optimization functions for each node and its neighbors. The local optimization function is then solved using Newton’s method of optimization instead of stochastic gradient descent, as it has a faster convergence rate [26]. The achievable service rate of each link is then calculated and its percentage error with the target service rate is formulated. In addition, the average throughput is finally calculated. The whole process is repeated under different network operational parameters as SINR thresholds, target service rates, and the number of nodes, to emphasize their effect on the average throughput. The main contribution of the presented work is giving a full description of the effect of changing network operational parameters on the average throughput and proving asymptotic relations using polynomial interpolation that describes the following. 

Throughput as a function of both target service rates and SINR threshold for a given number of nodes.Throughput as a function of both the number of nodes and the target service rates for a given SINR threshold.Maximum throughput for a different number of nodes at different SINR thresholds.

The proposed model proves its robustness against the increasing number of nodes due to its dependence on the local network size instead of the global one. The model also provides efficient asymptotic throughput relations to be used for estimating the performances of such wireless networks.

The rest of the paper is organized as follows. In Section 2, the system model is described, Section 2.1 describes the used CSMA scheduling algorithm and the research problem is explained. In Section 2.2, the global optimization function is introduced while the distributed local networks algorithm and its model are described in Section 2.3. In Section 2.4, the computational complexity is explained, Section 2.5 discusses the polynomial interpolation technique, and Section 2.6 analyzes the delay performance. In Section 3, the numerical analysis of the proposed model is obtained and described. Results are discussed in Section 4 and Section 5 concludes the work and results.

The used parameters and variables in the following sections and their descriptions are summarized and given in Table 1.

## 2. System Model

The model used in this article is based on a single-hop wireless network. Each node is formed from a pair of transmitter and receiver similar to the bipole model in [27], the distance between transmitter and receiver is $d_{ii}$ for node i. N is defined as the total number of nodes in the network model. Let $d_{ij}$ be the distance between two nodes i and j.

For scheduling data transmissions between nodes, x(t) is defined as the schedule of the network; it can also be referred to as x. In other words, $x_{i}\left(t\right)=1$ indicates that node i is active at time slot t and during data transmission. Two nodes are considered to be neighbors and interfere with each other if the distance between them is less than or equals to R (Close-in-Radius). Interference between nodes is neglected if they are not neighbors or, in other words, the distance between them is higher than R as shown in Figure 1.

The *SINR* model at node i as a function of x(t) is given by:(1)$$SINR=\frac{P_{i}d_{ii}{}^{-\alpha }}{\sum_{\left\{j\in \;N_{j}\;,\;j\neq I,\;x_{j}\left(t\right)=1\right\}}P_{j}\;d_{ji}{}^{-\alpha }\;+\omega }\;,$$
where $P_{j}$ is the transmitting power of node j, α is the path loss exponent of the standard path loss model ${\left|d\right|}^{-\alpha }$. ω is the variance of the Gaussian thermal noise present at all receivers. $N_{j}$ is a set of node j and its neighbors.

The condition for $x_{i}\left(t\right)=1$ (Node *i* is active and transmitting data) is SINR≥T; T is the *SINR* presumed threshold constraint. If all active nodes in a schedule satisfy this condition, their schedule is called a feasible schedule. The list of all feasible schedules is defined as I*,* where receivers are able to receive data successfully.

### 2.1. CSMA Scheduling Algorithm

The main aim of the CSMA scheduling algorithm is to ensure proper data packets reception. The procedure objective is obtaining the transmission attempt rate λ, hence supporting the required target service rates of each node for acquiring the optimal throughput. The scheduling algorithm is based on the SINR readings of the previous time slot, to ensure successful transmission of data if a certain threshold T is exceeded. The process is described in the following CSMA SINR threshold scheduling Algorithm 1.

**Algorithm 1.** CSMA SINR threshold scheduling algorithm-Each node is assigned with transmission attempt rate λ>0.-In each time slot, a randomly selected node i is allowed to update its schedule xi(t) based on the information in the previous time slot. **if**
SINRi(x(t−1))< T**, then**
xi(t)=0 and node i waits for another time slot to update its schedule again.
**else if**
$SINR_{i}\left(x\left(t-1\right)\right)\geq \;T$**, then**
Node i exchanges messages with neighbors, to find if they can meet their SINR requirements if link i gets activated. 
**if** any of its neighbors can’t meet its requirement, **then**
$x_{i}\left(t\right)=0$ and node i waits for another time slot to update its schedule again.
**else if** all neighbors can meet their SINR requirements if link i gets activated, **then**
$x_{i}\left(t\right)=1$ with probability $\frac{\lambda_{i}}{1+\lambda_{i}}$ and $x_{i}\left(t\right)=0$ with probability $\frac{1}{1+\lambda_{i}}$.
**end if**
**end if**

It can be shown in [25] that the adaptive CSMA algorithm induces a Markov chain on the state space of the schedules ${\left\{0,1\right\}}^{N}$. Further, the stationary distribution of the Markov chain, parametrized by the transmission attempt rate vector $\lambda \;=\;{\left[\lambda_{i}\right]}_{i=1}^{N}$, is given by:(2)$$p\left(x\right)=\frac{1}{z}\;\underset{j:x_{j}=1}{\prod }\lambda_{j}\;1\left(x\in I\;\right),\forall \;x\in {\left\{0,1\right\}}^{N}\;,$$
where 1(x ∈ I) is an indicator of x being a feasible schedule, and z is the normalizing constant. Then, due to the ergodicity of the Markov chain, the long-term service rate of a node i denoted by $s_{i}$ is equal to the marginal probability that node i is active, i.e., $p_{i}\left(x_{i}\;=\;1\right)$. Thus, the service rates and the transmission attempt rates are related as follows:(3)$$s_{i}=p_{i}\left(1\right)=\underset{x:\;x_{i}=1}{\sum }\frac{1}{z}\;\underset{j:x_{j}=1}{\prod }\lambda_{j},\forall \;i\in N\;,$$
where $p_{i}\left(1\right)$ denotes $p_{i}\left(x_{i}\;=\;1\right)$. The adaptive CSMA algorithm can support any service provided that appropriate transmission attempt rates are used for the underlying distribution [28]. If the desired service rates are known, these transmission attempt rates can be obtained by solving the system of equations in Equation (3).

Assume that each node i has a capacity of 1. If node i  transmits data all the time (without affecting other nodes), then its service rate is 1 (unit of data per unit time). Then, $s_{i}\left(r\right)$ is also the normalized service rate with respect to the node capacity.

The following concave function G(r) can be maximized by choosing a suitable value of the transmission aggressiveness r, this maximization is equivalent to the minimization of the Kullback–Leibler divergence between the arrival rate and the service rate distribution functions as established in [29].

### 2.2. The Global Optimization Function

The global optimization function of the proposed network model is given by:(4)$$r=\arg max_{\;r\in R^{N}}\;G\left(r\right)\;G\left(r\right)=\;\underset{k\in N}{\sum }s_{k}r_{k}-ln\left(\underset{y\in I\;}{\sum }exp\left(\underset{k\in N}{\sum }y_{k}r_{k}\right)\right)\;,$$
where r is the transmission aggressiveness vector of dimension N and it is a function of λ as r=ln(λ). let $y={\left[y_{k}\right]}_{k\in N}\;\in {\left\{0,1\right\}}^{N}$ be the global feasible schedule such that $y_{k}=0$ means that node k is inactive while $y_{k}=1$ indicates that node k is active and meets the required *SINR* threshold. The desired service rates vector is denoted as ${\left\{s_{i}\right\}}_{i\in N}$.

For a proper understanding of the global optimization Equation (4), let $\frac{\partial G\left(r\right)}{\partial r_{i}}=0$ to show that it solves Equation (3) and results in:(5)$$s_{i}=\;\frac{\sum_{y\in I:y_{i}=1}exp\left(\sum_{k=1}^{N}y_{k}r_{k}\right)}{\sum_{y\in I}exp\left(\sum_{k=1}^{N}y_{k}r_{k}\right)}\;\forall \;i\in N\;,$$

To solve Equation (4), the distributed stochastic gradient descent algorithm was used [23]. However, the gradient of (4) estimation was calculated in a distributed manner and took an impractical time of convergence in order to reach steady state. In order to rectify this problem, the proposed global optimization function was divided into separate and scalable approximated local optimization functions and, finally, these local functions were appropriately combined for estimating the solution to the global problem. The target service rates were assumed as predefined for each node.

### 2.3. The Local Optimization Function

The local optimization function was similarly structured as the global one with some parameters replaced by ones with a j index to represent its local attachment to node j and its neighbors. The local optimization function of node j was defined as:(6)$$r_{j}=argmax_{\;r\in R^{N_{j}}}\;F\left(r\right)\;F\left(r\right)=\underset{k\in N_{j}}{\sum }s_{k}r_{k}-ln\left(\underset{y\in I_{j}\;}{\sum }exp\left(\underset{k\in N_{j}}{\sum }y_{k}r_{k}\right)\right)\;,$$
where $r_{j}={\left[r_{jk}\right]}_{k\in \;N_{j}}$ is the local transmission aggressiveness of node j,$\;y={\left[y_{k}\right]}_{k\in N_{j}}\;\in {\left\{0,1\right\}}^{N_{j}}$ is the local feasible schedule at node j such that $y_{j}=0$ means that node j is inactive and $y_{j}=1$ indicates that node j is active and exceeds the *SINR* threshold. $s^{\left(j\right)}\;=\text{\{}s_{k}\;|k\in \;N_{j}\}$ is the local service rate vector of node j.

Due to the downscaling of the global network dimensions, the solution might be simplified by solving local optimization functions. The local solutions of the transmission attempt rate are then combined to produce a global transmission attempt rate that can be directly used in the CSMA algorithm. This process is chosen over the adaptation of transmission attempt rates using a stochastic gradient descent that requires extensive time to converge on the global function. Each node in the network executes (Algorithm 2) in parallel to get the average normalized throughput. 

**Algorithm 2.** Proposed Algorithm to obtain Average Normalized ThroughputInput: $\left(s_{k},\;k\in \;N_{j}\right)$; Output: Average Normalized Throughput $Th\left(s^{t}\right)$

(1)Neighbors of node j provide their target service rates.(2)Local optimization function (6) is solved using Newton’s method of optimization for node j with its surrounding neighbors and the maximum achievable service rates is locally obtained for all feasible schedules of the neighborhood.(3)Node j and each neighbor from $k\in \;N_{j}$ provide their locally maximum achievable service rates.(4)The average achievable service rate can be calculated for each node separately by averaging the service rates in each neighborhood contained in that node.(5)Calculate the approximate error between the achieved and the target service rates:(7)$$e\left(s^{t}\right)=\frac{\sum_{i=1}^{N}\left|s_{i}^{t}-s_{i}^{a}\right|}{N}\;,$$ where $s^{a}={\left[s_{i}^{a}\right]}_{i=1}^{N}$ are the service rates that can be achieved and $s^{t}={\left[s_{i}^{t}\right]}_{i=1}^{N}$ is the given target service rate vector.(6)Compute the average normalized throughput: (8)$$Th\left(s^{t}\right)=\frac{\sum_{i=1}^{N}s_{i}^{t}\;\left(1-e\left(s^{t}\right)\right)}{N}\;,$$(7)Use Polynomial interpolation to form the average normalized throughput equations as a function of either—number of nodes and target service rates or SINR threshold and target service rates. Polynomial interpolation is also used to get the maximum normalized throughput as a function of the number of nodes.


The approximate global transmission attempt rate $\tilde{\lambda }$ that can be directly applied to the CSMA SINR threshold scheduling algorithm (Algorithm 1) is given by:(9)$$\tilde{\lambda }\;={\left(\frac{1-s_{j}}{s_{j}}\right)}^{\left|N_{j}\right|-1}\underset{k\in N_{j}}{\prod }e^{r_{kj}}\;,$$
where $s_{j}$ is the target service rate of link j, $\left|N_{j}\right|$ is the number of nodes in the local neighborhood of node j. The transmission aggressiveness is $r_{kj},\;k\in \;N_{j}$ is the optimized parameter in Step (2) of node j and its presence in every local neighborhood. 

Newton’s method of optimization used in Step (2) is used to optimize the local optimization function (6) [30]. It can be computed without complexity due to its relatively smaller size as compared to the global function and is given by:(10)$$r^{\left(t\right)}=r^{\left(t-1\right)}-{\left([\nabla^{2}F\left(r^{\left(t-1\right)}\right){]}_{ik}{)}^{-1}\;.\;[\nabla F\left(r^{\left(t-1\right)}\right)\right]}_{k}\;,$$
where $r^{\left(t\right)}$ is the transmission aggressiveness of the current iteration and $r^{\left(t-1\right)}$ is of the previous iteration. ${\left[\nabla^{2}F\left(r^{\left(t-1\right)}\right)\right]}_{ik}$ and ${\left[\nabla F\left(r^{\left(t-1\right)}\right)\right]}_{k}$ are the Hessian matrix and the gradient vector of Newton’s method, respectively. The gradient and Hessian of node j computations are done through the distribution:(11)$${\hat{b}}_{j}\left(x^{\left(j\right)}\right)=\frac{1}{z_{j}}exp\left(\underset{k\in N_{j}}{\sum }x_{k}r_{k}\right),\forall \;x^{\left(j\right)}\in I_{j}\;,$$
where $z_{j}$ is the local normalization constant of node j and its neighbors. $I_{j}$ is a list of all feasible schedules of node j and its neighbors. The gradient of (6) is given by:(12)$${\left[\nabla F\left(r\right)\right]}_{k}=s_{k}-m_{k}\left(r\right),\;k\in N_{j}\;,$$
where $m_{k}\left(r\right)=p\left(x_{k}=1\right)$ under ${\hat{b}}_{j}$ distribution for $k\in N_{j}$.

The Hessian of (6) is given by:(13)$${\left[\nabla^{2}F\left(r\right)\right]}_{ik}=\{\begin{array}{c} m_{i}\left(r\right)m_{k}\left(r\right)-m_{ik}\left(r\right),\;i,k\in N_{j},i\neq j\\ m_{k}{\left(r\right)}^{2}-m_{k}\left(r\right),\;i=k. \end{array}\;,$$
where $m_{ik}\left(r\right)=p\left(x_{i}=1,x_{k}=1\right)$ of the distribution ${\hat{b}}_{j}$.

### 2.4. Computational Complexity

It is possible to implement the Newton method in (Algorithm 2). This is because it is possible to analytically calculate the gradient and the Hessian of the local objective function F(r) since the problem dimension is reduced from the global one.

The gradient and the Hessian calculations need information about the feasible local schedules at a link. This information is specifically needed to calculate the normalization constant z. These computations are feasible because of the complexity of $O\left(2^{\left|N_{j}\right|}\right)$ associated with a computation scale of only a particular size of the local neighborhood that is independent of the network’s global size, which could be considerably large.

### 2.5. Polynomial Interpolation

The polynomial cubic interpolation used in Step (7) is explained in [31,32,33] and has a common form of:(14)$$f\left(\theta \right)=a_{0}+a_{1}\theta +a_{2}\theta^{2}+\ldots +a_{n}\theta^{n},\;a_{n}\neq 0\;,$$
where θ is the variable of the function f and n is the degree of the polynomial. 

The solution of Equation (14) is computed using the Vandermonde matrix [33] formed in Equation (15), in order to calculate the coefficient $a_{n}$ given both θ and f:(15)$$\left[\begin{array}{ccccc} 1 & \theta_{1} & \theta_{1}^{2} & \cdots & \theta_{1}^{m}\\ 1 & \theta_{2} & \theta_{2}^{2} & \cdots & \theta_{2}^{m}\\ 1 & \theta_{3} & \theta_{3}^{2} & \cdots & \theta_{3}^{m}\\ \vdots & \vdots & \vdots & \ddots & \vdots \\ 1 & \theta_{n} & \theta_{n}^{2} & \cdots & \theta_{n}^{m} \end{array}\right]\left[\begin{array}{c} a_{0}\\ a_{1}\\ a_{2}\\ \vdots \\ a_{m} \end{array}\right]=\left[\begin{array}{c} f_{0}\\ f_{1}\\ f_{2}\\ \vdots \\ f_{n} \end{array}\right]\;,$$

Another form of the polynomial interpolation used in this work is the bicubic interpolation for a function of two variables, such as:(16)$$f\left(\theta ,\varphi \right)=a_{0}+a_{1}\theta +a_{2}\varphi +a_{3}\theta^{2}+a_{4}\theta \varphi +a_{5}\varphi^{2}+a_{6}\theta^{3}+a_{7}\theta^{2}\varphi +a_{8}\theta \varphi^{2}+a_{9}\varphi^{3},$$
where θ and φ are the two variables of the function f.

The solution of Equation (16) is similar to Equation (14) and is computed using the Vandermonde matrix formed in Equation (17) to calculate the coefficient $a_{n}$ given θ, φ, and f:(17)$$\left[\begin{array}{ccccccc} 1 & \theta_{1} & \varphi_{1} & \theta_{1}^{2} & \theta_{1}\varphi_{1} & \cdots & \varphi_{1}^{m}\\ 1 & \theta_{2} & \varphi_{2} & \theta_{2}^{2} & \theta_{2}\varphi_{2} & \cdots & \varphi_{2}^{m}\\ 1 & \theta_{3} & \varphi_{3} & \theta_{3}^{2} & \theta_{3}\varphi_{3} & \cdots & \varphi_{3}^{m}\\ \vdots & \vdots & \vdots & \vdots & \vdots & \ddots & \vdots \\ 1 & \theta_{n} & \varphi_{n} & \theta_{n}^{2} & \theta_{n}\varphi_{n} & \cdots & \varphi_{n}^{m} \end{array}\right]\left[\begin{array}{c} a_{0}\\ a_{1}\\ a_{2}\\ \vdots \\ a_{m} \end{array}\right]=\left[\begin{array}{c} f_{0}\\ f_{1}\\ f_{2}\\ \vdots \\ f_{n} \end{array}\right]\;,$$

Equations (15) and (17) are solved using the Gaussian elimination method [34] until a reduced echelon form is reached; hence, the values of the coefficients are computed. In Step (7), Polynomial interpolation is used to obtain the average normalized throughput as a function of either number of nodes and target service rates or *SINR* threshold and target service rates. Polynomial interpolation is also used to get the maximum normalized throughput as a function of the number of nodes.

### 2.6. Delay Performance

Considering the delay performance, it is known that the CSMA Markov chain’s impractically slow mixing time leads to a bad delay performance [28,35]. Recent work such as [36,37] have enhanced delay performance by using several parallel CSMA Markov chain cases. These results [36] are demonstrated on the assumption that the ideal transmission attempt rate is pre-computed and is easily accessible to the algorithm. The proposed local algorithm can be used in combination with the methods in [36,37] to achieve a practical CSMA algorithm with an excellent throughput and a low delay, to effectively estimate these transmission attempt rates.

## 3. Results

In this section numerical analysis are used to estimate the performance of the proposed algorithm. Random topology graphs are considered with a different number of nodes to test the robustness of the model. Random networks are generated by placing nodes on a two-dimensional area of length of 12 unit distance. The system parameters are summarized in Table 2.

Random networks are generated with varying densities from 10 up to 100 nodes, and vertices are drawn when nodes are neighbors and interfere with each other. The total number of links is higher than the total number of nodes, but it does not reach the mesh topology where the number of links equals to N(N−1)/2. The interference graph of a 100-node random topology is shown in Figure 2.

In order to prove the fast convergence rate of the proposed model, Figure 3 shows that the norm of the gradient in Newton’s method of a random local network sample converges in 4 to 5 iterations.

In Figure 4, the average normalized throughput as a function of the target service rate is shown for different network topologies of 10, 30, 50, and 100 randomly distributed nodes at 9, 12 and 15 dB SINR thresholds.

As shown in Figure 4, increasing the SINR threshold shows degradation in throughput, especially at high-target service rates. This is because the nodes will be unable to transmit until a higher SINR threshold is met. Meanwhile, it is observed that initially, as target service rate increases, the average normalized throughput increases until it reaches its maximum. This is where the network works under “stable operating conditions”. After that point, the network enters “unstable operating conditions” in which collisions become more likely and the number of backlogged frames increases. This means that the arrival rate of new frames to the system will be larger than the capability of successful frames transmission, thus, leading to a decrease of the average normalized throughput.

In order to acquire the obtained results in Figure 4 as a system of asymptotic relations, polynomial bicubic interpolation algorithm is used. Therefore, the average normalized throughput can be estimated via the following proposed equation in the general form of:(18)$$Th\left(N,S\right)=a_{0}+a_{1}N+a_{2}S+a_{3}N^{2}+a_{4}NS+a_{5}S^{2}+a_{6}N^{3}+a_{7}N^{2}S+a_{8}NS^{2}+a_{9}\;S^{3},$$
where Th is the average normalized throughput as a function of the number of nodes N and target service rate is denoted as S. The constants $a_{0},\ldots ,\;a_{9}$ are the constant coefficients in Equation (18) and are given in Table 3.

The interpolation equations are used to describe the missing values between the calculated throughput results and also give more details in three-dimensional planes of the average normalized throughput as a function of both target service rates and the number of nodes. In Figure 5, the plane is represented at different SINR thresholds of 9 up to 15 dB. It can be observed that increasing the number of transmitting nodes in the network will cause more collisions, thus, leading to a decrease in the network throughput.

Another way to make use of the polynomial bicubic interpolation is to generate another form of the previous equation with different parameters, such as:(19)$$Th\left(T,S\right)=c_{0}+c_{1}T+c_{2}S+c_{3}T^{2}+c_{4}\;TS+c_{5}S^{2}+c_{6}T^{3}+c_{7}T^{2}S+c_{8}TS^{2}+c_{9}\;S^{3},$$
where Th is the average normalized throughput as a function of target service rate denoted as S and the *SINR* threshold T. The constants $c_{0},\ldots ,\;c_{9}$ are the constant coefficients in Equation (19) and are given in Table 4.

Moreover, the previous step of interpolation is repeated to find an asymptotic relation of the average normalized throughput as a function of both target service rates and the SINR threshold. In Figure 6, the plane is shown with a number of nodes from 10 up to 100 nodes.

After obtaining the average normalized throughput from the numerical analysis, the maximum normalized throughput could be extracted. Polynomial cubic interpolation algorithm was used to concatenate the above figures and define a relation for acquiring the maximum normalized throughput in the general form of:(20)$$Th\_max\left(N\right)=k_{0}+k_{1}N+k_{2}N^{2},$$
where Th_max is the maximum normalized throughput as a function of the number of nodes *N*. The constants $k_{0},\;k_{1},\;\mathrm{and}\;k_{2}$ are the constant coefficients in Equation (20) and are given in Table 5.

The interpolation equations graphs of the maximum normalized throughput are shown in Figure 7 as a function of the number of nodes at the different SINR thresholds from 9 up to 15 dB.

Another way to describe the maximum normalized throughput more generally, using the polynomial bicubic interpolation is to formulate a plane that is a function of, both, the number of nodes and the SINR threshold. Therefore, the maximum normalized throughput was estimated using the following equation as the general form:(21)$$Th\_max\left(N,T\right)=e_{0}+e_{1}N+e_{2}T+e_{3}N^{2}+e_{4}NT+e_{5}T^{2},$$
where Th_max is the maximum normalized throughput as a function of the number of nodes N and the SINR threshold denoted as T. The constants $e_{0},\ldots ,\;e_{5}$ are the constant coefficients in Equation (21) and are given in Table 6.

As a result to Equation (21), the maximum normalized throughput as a function of both the number of nodes and the SINR threshold can be generated approximately. Different from Equation (20), the new equation adds the dimension of the SINR threshold to give a more general description and eases the prediction to be based on two parameters instead of one. Equation (21) is shown in Figure 8 for different number of nodes and SINR thresholds.

A sample of error using Equation (7) between the proposed approximation technique and the stochastic gradient descent algorithm is shown in Figure 9. Where the local approximation algorithm that calculates the approximate transmission attempt rates uses their static values in the CSMA algorithm (i.e., they are not adapted during the algorithm). On the other hand, the stochastic gradient descent algorithm begins with some initial transmission attempt rates, and by observing the corresponding service rates, it adapts the transmission attempt rates [23].

For 5G networks, the error has to be minimum at all target service rates. Additionally, determining the transmission attempt rate is an NP-hard problem and the stochastic gradient descent is unpractical as it adapts those transmission attempt rates in several iterations, as shown in the Figure 9. This is why the proposed approach of pre-calculating those transmission attempt rates and using them directly in the network helps in reducing the error at most target service rates. The results would be comparable even at higher rates but there still exists an advantage in the early stages where the network works under “stable operating conditions”. The error of the proposed model did not exceed 0.06, while the gradient descent started with an error exceeding 0.2 for a target service rate up to 0.4 unit data per unit time.

## 4. Discussion

The proposed model aims at providing possible ways of predicting random network performances under different circumstances. One thing to mention is that the proposed model is scalable, as it studied a 10- to 100-node random topology in an area of 12 × 12 square units of distance. This means that if a 400-node random topology is studied at 24 × 24 square units of distance, the performance will be similar to the proposed 100-node one. This is due to the dependence of the proposed model on the local neighborhood size, not the global one. Therefore, what matters is the performance degradation that would arise from increasing the number of nodes in the same limited area. Additionally, increasing the given area can allow for fitting a higher number of nodes, as the limited area considered in this work affects the interference to a great extent, due to the high-density neighborhoods interfering with each other. Additionally, increasing the number of nodes in the given area to a value higher than the studied ones might lead to a density (number of nodes/unit area)>1, which is rarely found and should most of the time be ≤1. Additionally, the resulting model might be used for scalable networks. In other words, the resulted network could be used to estimate the performance of both small as well as large networks. The presented analysis is based on the normalized unit area and the normalized throughput, so it might provide a good performance whatever be the network size.

The proposed polynomial interpolation’s throughput asymptotic relations here are based on the studied topologies and might lead to other performance prediction if different parameters were used. To sum up the used parameters here, the SINR threshold was set to 9, 12, and 15 dB, up to 0.9 unit target service rates, and up to 100 nodes randomly distributed in the unit area were considered.

Three-dimensional topologies were applicable and would not have differed much from the two-dimensional topology used in the proposed model, as the distance was the only matter. We previously mentioned the close-in-radius R distance as the distance where interference was neglected if two nodes were at distance>R apart from each other. This radius could either be of a circle if a two-dimensional topology was considered or a radius of a sphere in case of the three-dimensional topology. Therefore, at the end what matters is distances at any directions and the three-dimensional topology could be equivalent to a dense network that is already studied in the proposed model. Additionally, the main scope of this paper was to calculate the effect of the increasing the number of nodes on the throughput in dense networks, which is suitable for 5G applications. The energy and the delay could be considered in details in future work.

## 5. Conclusions

A CSMA algorithm of a single-hop wireless network was considered under a realistic SINR model. An approximation algorithm of distributing the global network into downscaled local neighborhoods was used to calculate the transmission attempt rate to optimize the throughput of the global network. This was done by achieving target service rates of nodes while varying the number of nodes up to a random 100-node topology. The proposed model converged fast and proved its robustness against the increasing number of nodes as it depended only on the size of the local network instead of the global dense one. Three-dimensional planes of the average normalized throughput were obtained by polynomial interpolation that produced a complete description of the performance of the network. Maximum normalized throughput was obtained too as a function of the number of nodes using polynomial interpolation. The used approach of pre-calculating the transmission attempt rates and using them directly in the network helped in reducing the error at most values of the target service rates. Even at higher rates, results would be comparable with a gradient descent algorithm but there still would exist an advantage in the early stages where the network would work under “stable operating conditions”. The proposed model is also scalable, so it might provide a network performance, whatever be its size. Additionally, the proposed model has a faster convergence time and is considered to be less complex and more practical than the previously used gradient descent algorithm.

## Figures and Tables

**Figure 1 sensors-19-03651-f001:**
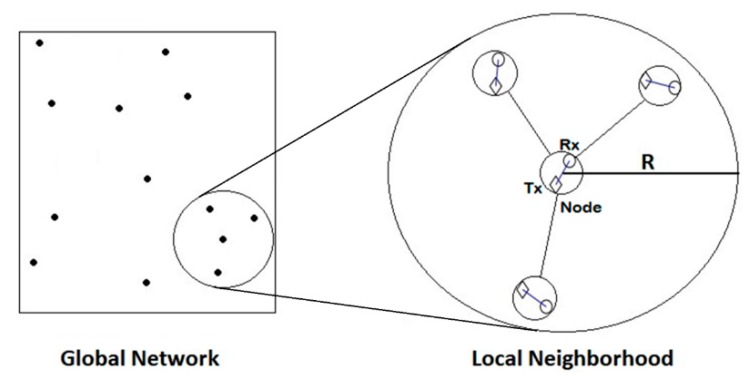
Local Neighborhood network as a part of the global network, the maximum distance between a node (a pair of transmitter and receiver) and its neighbors is the Close-in-Radius distance R

**Figure 2 sensors-19-03651-f002:**
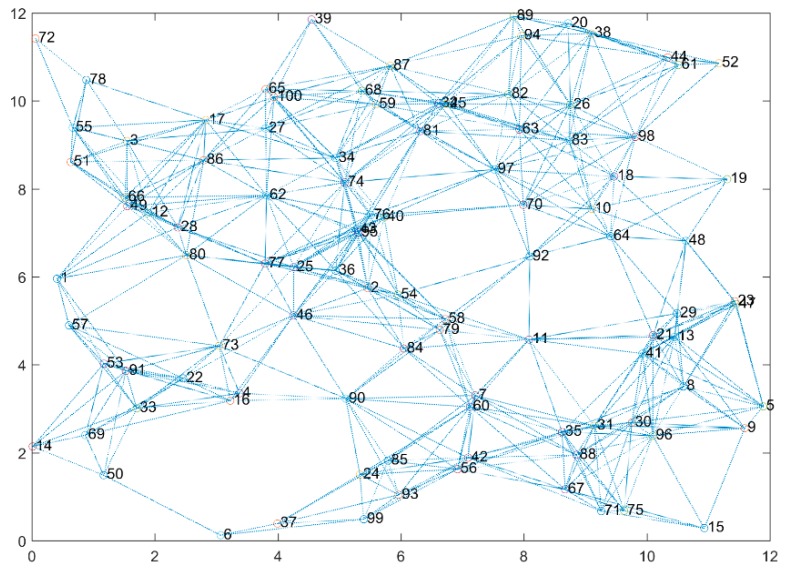
Interference graphs of 100-node random topology.

**Figure 3 sensors-19-03651-f003:**
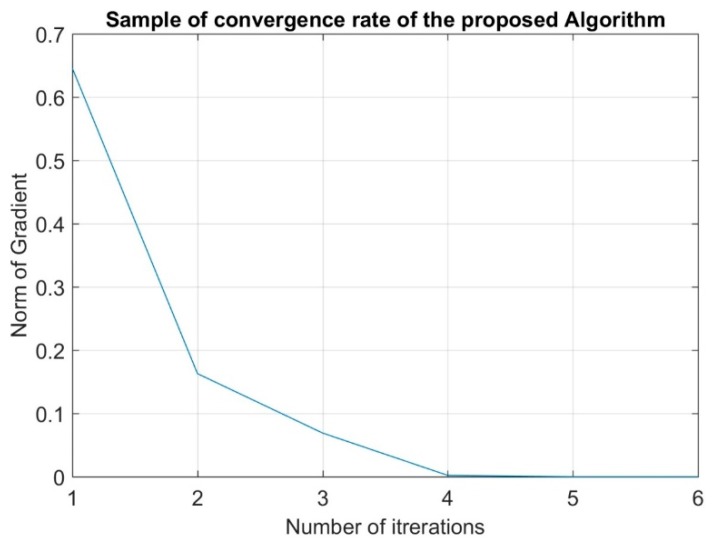
Sample of Convergence Rate of the proposed algorithm.

**Figure 4 sensors-19-03651-f004:**
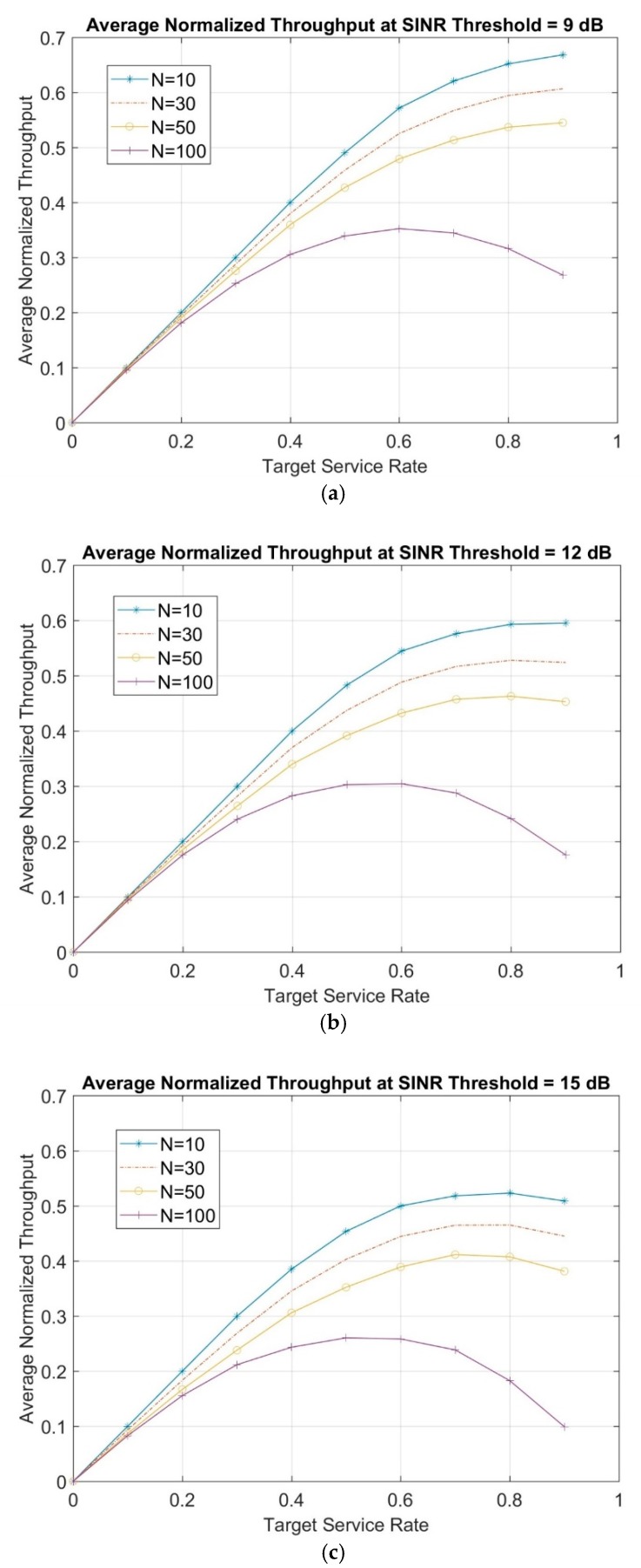
Average normalized throughput for a different number of nodes—(**a**) 9 dB SINR threshold, (**b**) 12 dB SINR threshold, and (**c**) 15 dB SINR threshold.

**Figure 5 sensors-19-03651-f005:**
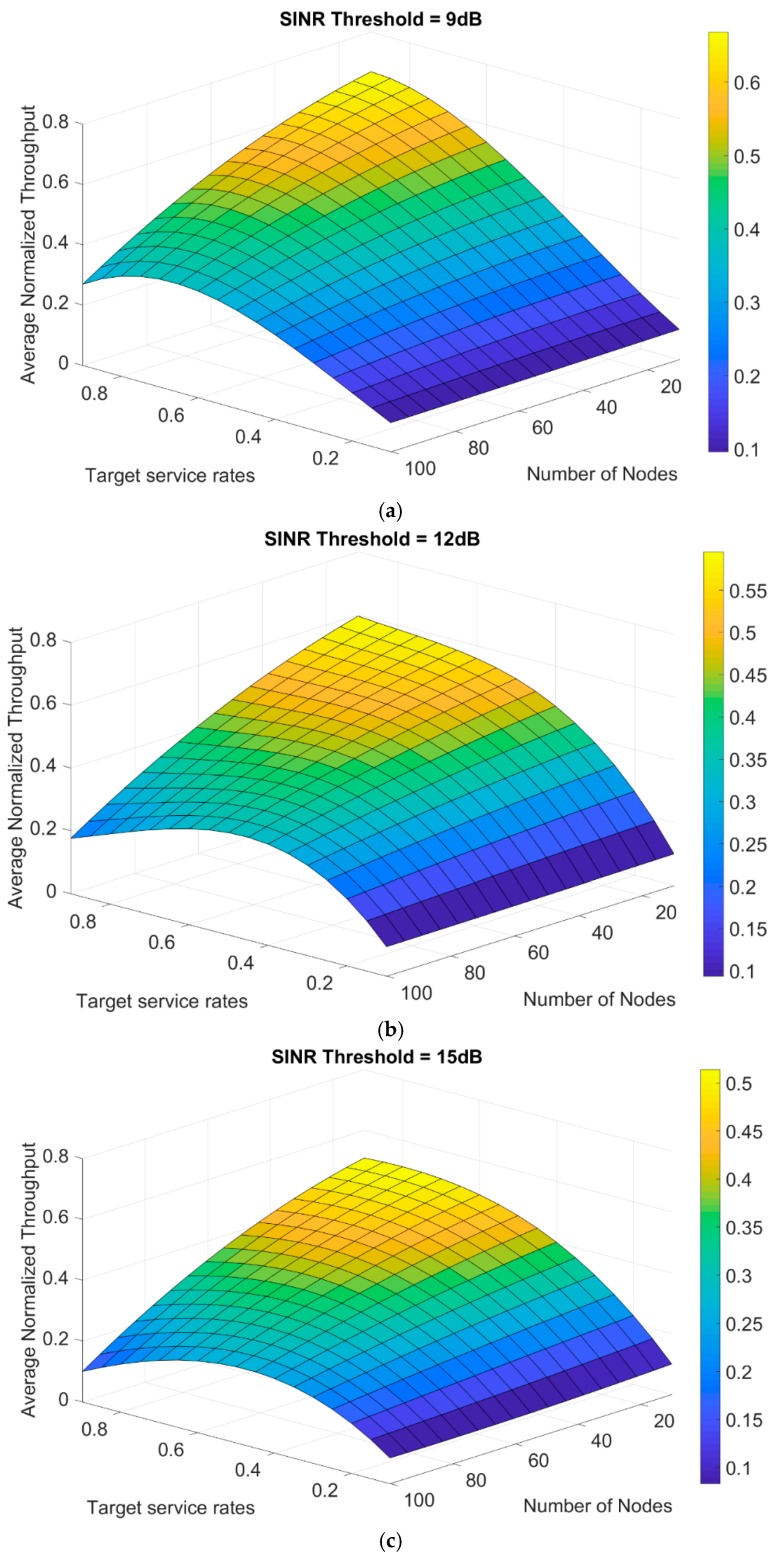
Plane of average normalized throughput generated from polynomial interpolation at a different number of nodes for—(**a**) 9 dB SINR threshold, (**b**) 12 dB SINR threshold, and (**c**) 15 dB SINR threshold.

**Figure 6 sensors-19-03651-f006:**
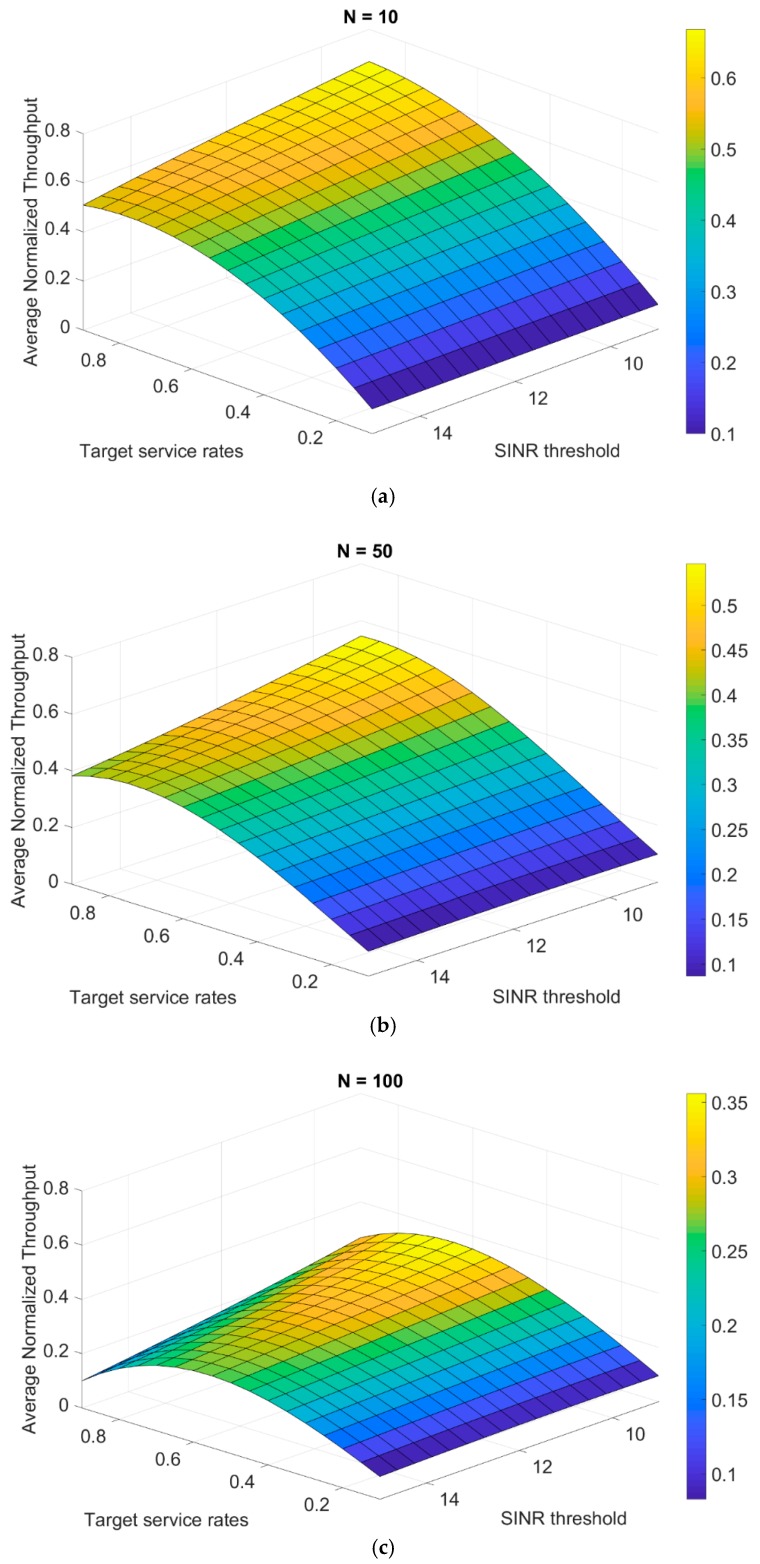
Plane of average normalized throughput generated from the polynomial interpolation for different SINR thresholds of—(**a**) 10-node topology, (**b**) 50-node topology, and (**c**) 100-node topology.

**Figure 7 sensors-19-03651-f007:**
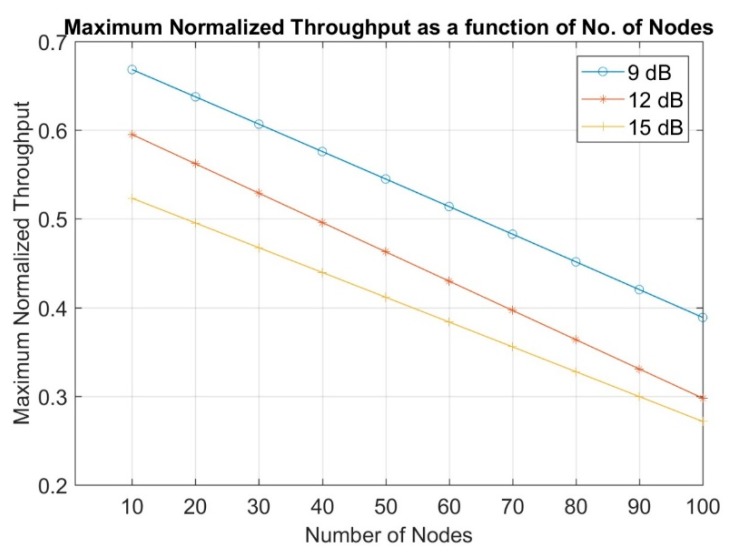
Maximum normalized throughput at different number of nodes for SINR threshold of 9 dB, 12 dB, and 15 dB.

**Figure 8 sensors-19-03651-f008:**
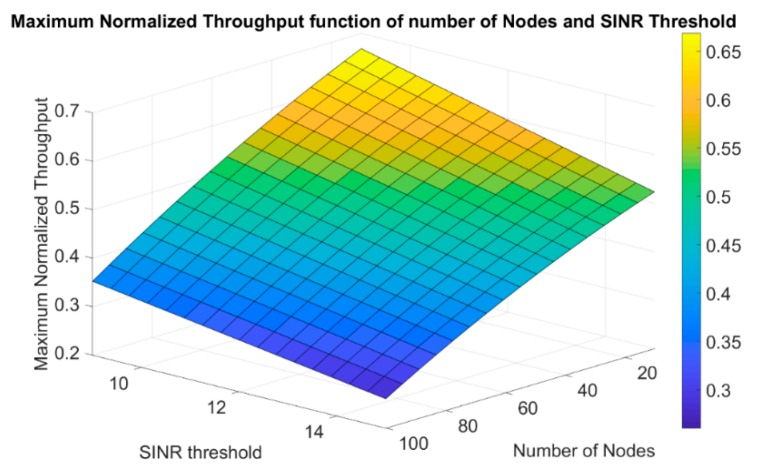
Maximum normalized throughput as a function of both the number of nodes and the SINR threshold.

**Figure 9 sensors-19-03651-f009:**
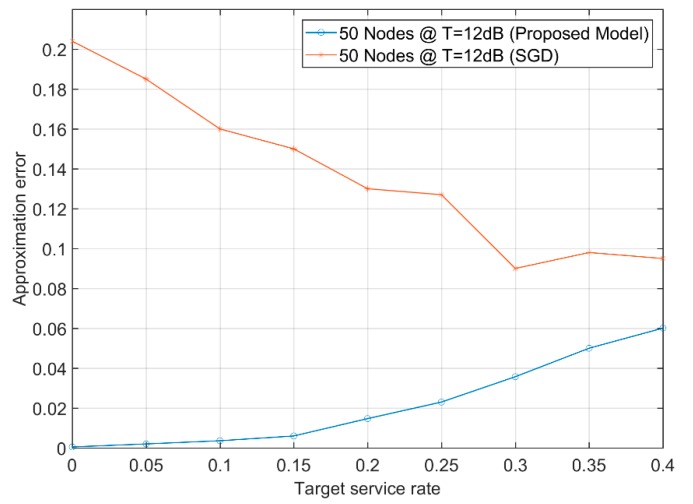
The approximate error of the achievable and the target service rates due to stochastic gradient descent and the proposed approximation algorithm with 50 nodes distributed with the same random topology used in this article.

**Table 1 sensors-19-03651-t001:** System parameters and their descriptions.

Parameter/Variable	Description
$d_{ii}$	Distance between transmitter and receiver of same node i.
$d_{ij}$	Distance between two nodes i and j.
N	Total number of nodes.
$N_{j}$	Number of nodes at node j neighborhood.
x(t)	Schedule of the network where $x_{i}\left(t\right)=1$ means that link i is active and transmitting data and $x_{i}\left(t\right)=0$ means that link i is not active.
R	Close-in-Radius distance where interference is neglected if distances between nodes exceeded it.
SINR	Signal to Interference and Noise Ratio.
$p_{i}$	Transmit power of link i.
α	Path loss exponent.
ω	The variance of the Gaussian thermal noise present at all receivers.
T	SINR threshold that has to be exceeded to ensure successful data reception.
I	List of all feasible schedules.
λ	Transmission attempt rate
$s_{i}$	The long-term service rate of node i which is the marginal probability that node i is active.
1(x∈I)	The indicator for the feasibility of the schedule.
z	Normalizing constant.
r	Transmission aggressiveness.
$y_{k}$	The feasible schedule such that $y_{k}=0$ means that node k is inactive while $y_{k}=1$ indicates that node k is active and meets the required SINR threshold.
Th	Average Normalized Throughput.
Th_max	Maximum Normalized Throughput
$e\left(s^{t}\right)$	The approximate error between the achieved and the target service rates.
P(x)	the stationary distribution of the CSMA Markov chain

**Table 2 sensors-19-03651-t002:** The system parameters used in numerical analysis.

System Parameter	Value
Dimensions of the interference graph (unit area)	12 × 12
Distance between transmitter and its corresponding receiver (unit distance)	0.5
Path loss exponent	3
Close in Radius (unit distance)	2.5
SINR threshold (dB)	9 up to 15
Target service rate (Unit of data per unit time)	0 up to 0.9
Transmit power (unit power)	1

**Table 3 sensors-19-03651-t003:** Coefficients of Equation (18) for different SINR thresholds.

SINR Threshold	$a_{0}$	$a_{1}$	$a_{2}$	$a_{3}$	$a_{4}$	$a_{5}$	$a_{6}$	$a_{7}$	$a_{8}$	$a_{9}$
T=9	0.3476	$-3\times 10^{-4}$	0.559	$3.5\times 10^{-6}$	0.0029	1.096	$10^{-9}$	$-3.4\times 10^{-5}$	−0.00466	−1.016
T=10	0.1716	$-4.6\times 10^{-4}$	−1.2147	$3.4\times 10^{-6}$	0.004496	5.442	$10^{-9}$	$-3.26\times 10^{-5}$	−0.0064	−3.865
T=11	−0.314	$3.65\times 10^{-4}$	4.92	$3.3\times 10^{-6}$	−0.00486	−8.14	$10^{-9}$	$-3\times 10^{-5}$	0.00256	4.28
T=12	−0.119	$8.45\times 10^{-5}$	2.49	$3.18\times 10^{-6}$	−0.00189	−2.97	$10^{-9}$	$2.8\times 10^{-5}$	$7.5\times 10^{-4}$	1.239
T=13	−0.305	$3.65\times 10^{-4}$	4.896	$4\times 10^{-6}$	−0.00614	−8.66	$10^{-9}$	$-3\times 10^{-5}$	0.0038	4.81
T=14	−0.3	$1.9\times 10^{-4}$	4.898	$5.3\times 10^{-6}$	−0.0059	−8.73	$10^{-9}$	$-3.4\times 10^{-5}$	0.004	4.847
T=15	−0.0515	$-4.25\times 10^{-4}$	1.718	$6.6\times 10^{-6}$	$-7.35\times 10^{-4}$	−1.597	$10^{-9}$	$-3.76\times 10^{-4}$	$-5.79\times 10^{-4}$	0.447

**Table 4 sensors-19-03651-t004:** Coefficients of Equation (19) for different number of nodes.

No. of Nodes	$c_{0}$	$c_{1}$	$c_{2}$	$c_{3}$	$c_{4}$	$c_{5}$	$c_{6}$	$c_{7}$	$c_{8}$	$c_{9}$
N=10	−0.028	−0.00138	1.32	$8.95\times 10^{-5}$	0.0166	−0.315	$10^{-6}$	$8.95\times 10^{-4}$	−0.028	−0.128
N=20	−0.0367	0.0025	1.25	$-1\times 10^{-4}$	0.0015	−0.112	$10^{-6}$	$-1.66\times 10^{-4}$	−0.03	−0.219
N=30	−0.069	0.0063	1.5	$-2.89\times 10^{-4}$	−0.0135	−0.566	$10^{-7}$	$-5.66\times 10^{-4}$	−0.0324	0.0755
N=40	−0.077	0.01069	1.43	$-4.8\times 10^{-4}$	−0.0328	−0.313	$10^{-7}$	0.00129	−0.03	−0.0665
N=50	−0.449	0.01425	0.8649	$-6.74\times 10^{-4}$	−0.0437	0.9377	$10^{-8}$	0.002	−0.0365	−0.752
N=60	−0.0276	0.0143	0.64	$-6.76\times 10^{-4}$	−0.0428	1.339	$10^{-8}$	0.00197	−0.036	−1.025
N=70	−0.0265	0.0143	0.623	$-6.79\times 10^{-4}$	−0.0419	1.29	$10^{-8}$	0.0019	−0.036	−1.04
N=80	−0.134	0.0165	1.942	$-6.8\times 10^{-4}$	−0.0647	−1.38	$10^{-8}$	0.00185	−0.012	0.366
N=90	−0.0599	0.0156	1.006	$-6.8\times 10^{-4}$	−0.052	0.516	$10^{-9}$	0.0018	−0.023	−0.77
N=100	−0.061	0.0156	1.019	$-6.86\times 10^{-4}$	0.0517	0.409	$10^{-9}$	0.00175	−0.0229	−0.749

**Table 5 sensors-19-03651-t005:** Coefficients of Equation (20) for different SINR thresholds.

SINR Threshold	$k_{0}$	$k_{1}$	$k_{2}$
S=9	0.6989	−0.003	$-4\times 10^{-7}$
S=12	0.6282	0.0033	$-7.48\times 10^{-8}$
S=15	0.551	−0.00278	$-1.2\times 10^{-7}$

**Table 6 sensors-19-03651-t006:** Coefficients of Equation (21).

Coefficient	$e_{0}$	$e_{1}$	$e_{2}$	$e_{3}\;$	$e_{4}\;$	$e_{5}\;$
Value	0.9213	−0.00345	−0.025	$\;-8.47\times 10^{-6}$	$9.81\times 10^{-5}$	$10^{-5}$

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
