# Peer review of "Aggregated Throughput Prediction for Collated Massive Machine-Type Communications in 5G Wireless Networks"

_sensors, 2019, doi:10.3390/s19173651_

Round 1

Reviewer 1 Report

The article is well written and presents an interesting topic in 5G. It is easy to read and follow. I just have some minor comments about the evaluation:

- First of all by those types of random topology graphs were chosen, why with those sizes and number of nodes? It would be nice to have some references of "typical" topologies and size to understand if the simulations are meaningful for 5G scenarios or not so much.

- Why the topologies are two-dimensional? I understand it is because of simplicity, but it would be nice to check whether the authors have tried three-dimensional topologies and if differences appear or not?

- How many runs per simulation are performed? The authors show the average results, but it would be nice to represent the standard deviation as well.

- Have the authors tried with topologies with more than 100 nodes? It seems than from 100 nodes on, the network starts to "degrade" somehow. Is it so? What is the biggest node density that their algorithm allows?

- Finally, would it be possible to have a comparison with other approaches? The related work is explained well in the introduction, but some comparison (even qualitative) at the end would be nice.

Additionally, there are some typos, like "previuos" in the article, so the authors should re-read it if possible, although in general it is well-written.

And I think Figures 4, 5, 6, 7, 8, 9 are too big in my opinion. I would suggest to reduce them if possible and, if feasible, have Figure 5, 6 and 7 in a single page (currently the use more than one page each and it is slightly confusing).

Reviewer 2 Report

The approach to attack the problem is quite interesting, but the model is simplistic and is evaluated from the network point of view; however, KPIs such as energy consumption of the nodes and delay in the access attempts are avoided. 

Figure 2 should be improved for the sake of clarity; using the algorithm environment is suggested. Also, several variables and parameters are used, a table summarizing those will be useful.

Various grammar and typo errors can be found along the text which makes difficult to follow and understand some of the ideas, sentences, and explanations given.

Reviewer 3 Report

This paper presented an approximation technique to overcome large network sizes, and utilized Newton’s method to obtain optimal throughput. The polynomial interpolation was used to estimate the average normalized throughput, and three-dimensional planes of the average throughput were presented that produced a complete description of the performance of the network. However, the presentation of this work requires substantial improvement. Furthermore, there are some technical problems that need to be addressed. Detailed comments are given as follows.

1. The format of the equations needs to be uniform. For example, there is a comma after equation (2) and no comma after equation (9). In addition, the label of equation (8) is not on the same line as the equation. The authors are suggested to revise the corresponding parts.

2. The global optimization function (3) is confusing, the reason that the desired service rates vector can be denoted as s_i should be explained in more detail.

3. The authors said that “Due to the downscaling of the global network dimensions, the solution may be simplified by solving local optimization functions”, but why the local solutions can be combined to produce a global transmission attempt rate?

4. This paper covers a number of variables, and it is better to give a variable scale for the sake of reference.

5. The description of some figures, such as figure 5 and figure 6, in this paper is limited to the trend of the curves, the reason behind should be added.

6. This paper covers a number of variables, and it is better to give a variable scale for the sake of reference.

7. There are many formatting errors in the references, and the paper needs a thorough revision before final submission. For example, the reference 3 and 15 are both published in IEEE/ACM Transactions on Networking, but one is added the “(ToN)”, another is not.

8. Some related works are missing, and they have to be added, e.g.,

https://ieeexplore.ieee.org/document/8660405

https://ieeexplore.ieee.org/document/6201210

https://ieeexplore.ieee.org/document/7317725/

https://ieeexplore.ieee.org/document/8067685

9. The difference between x and y should be introduced, which all mean the activation of node.

10. It is recommended that the simulation results have more quantitative analysis. The unit of rate should be labeled in figures for easy understanding.

11. In the page 1, there lacks a full name of “mMTC” at it first appearances.

12. In Abstract, the authors said the algorithm have lower complexity, so the authors should explain the computational complexity of the proposed algorithm in detailed to prove that the complexity is indeed reduced.

13.The reference format is not uniform. The journal formats of some of the references are not in abbreviated form.

14. The establishment of formula (3) requires more explanation.

15. There are various grammar mistakes and typos throughout this paper. Some examples are summarized as follows:

1) Why are "nodes" in "the number of nodes" lowercase in Abstract and capitalized in other places.

2) In the page 16. “…the maximum normalized throughput in a general from of:”. The word “from” doesn’t seem to fit this context, and consider replacing it with “form”.

Reviewer 4 Report

The manuscript presents an interesting approach to predict the large amount of sensors connected in 5G. Although you presented a large number of results, you didn't compare your method with state of the art techniques. This is required to observe the advantage of your mechanism.

Round 2

Reviewer 1 Report

The authors have improved a lot the article and it looks pretty neat. Currently, only minor English typos should be checked, like "resulted model" that should be "resulting model". 

Reviewer 2 Report

The authors have successfully addressed the first-round review comments.

Table 1 (Section 2.7) is useful as a reference to find quickly and present the parameters/variables that will be used along the paper; therefore, it should be introduced at the end of Section 1.

In figures 6,7, and 9 please add the colorbar for better interpretation of the plots.

Please proofread and fix minor grammar and typo errors.

Reviewer 3 Report

I am satisfied with the authors' revision.

Author Response

We would like to register our deep thanks for your sincere efforts and professional review of the submitted manuscript entitled “Aggregated Throughput Prediction for Collated Massive-Machine Type Communications in 5G Wireless Networks”.